# Qualitative and Arts-Based Evidence from Children Participating in a Pilot Randomised Controlled Study of School-Based Arts Therapies [note 1]

**DOI:** 10.3390/children9060890

**Published:** 2022-06-15

**Authors:** Zoe Moula, Joanne Powell, Vicky Karkou

**Affiliations:** 1Faculty of Medicine, Department of Public Health & Primary Care, Imperial College London, London SW7 2AZ, UK; 2Research Centre for Arts & Well-Being, Edge Hill University, Ormskirk L39 4QP, UK; powellj@edgehill.ac.uk (J.P.); karkouv@edgehill.ac.uk (V.K.)

**Keywords:** art therapy, music therapy, dance movement therapy, dramatherapy, children, schools, randomised controlled study (RCT), mental health, well-being, prevention

## Abstract

(1) Background: There is limited evidence on the impact of arts therapies as a tool for the prevention of mental health difficulties in childhood. This pilot randomised controlled study aimed to investigate the impact of arts therapies on children’s mental health and well-being; the qualitative and arts-based evidence is presented in this article. (2) Methods: Sixty-two children (aged 7–10) with mild emotional and behavioral difficulties were recruited across four primary schools and were randomly assigned to either art therapy, music therapy, dance movement therapy, or dramatherapy. All children were interviewed individually after their participation in arts therapies. (3) Results: Children verbally and artistically expressed that they experienced positive changes in their mental health and well-being, such as improved self-expression, safety, empowerment, hope, and optimism for the future. The arts were particularly important for expressing complex emotions and feelings that cannot be easily verbalised. Recommendations are provided to improve the quality of group arts therapies in future interventions, such as through smaller groups, longer sessions, and strategies to protect the therapeutic environment. (4) Conclusions: This study embraced all arts therapies as one research domain and set children’s verbal and non-verbal responses at the heart of outcome evaluation. This article highlights the importance of incorporating qualitative and arts-based methods to capture changes in children’s mental health well-being in future experimental studies.

## 1. Introduction

Children’s mental health and well-being has become central to global policy, such as the United Nations Sustainable Development Goals for Good Health and Well-Being [1] and the World Health Organization [2] call for the integration of mental health provision within the school curriculum. Despite that, the National Healthcare System (NHS) in the UK reported an increase in mental health disorders among children and adolescents from 1 in 9 in 2017 to 1 in 6 in 2020, with a sharp rise in self-harm, eating disorders, sleep disturbance, depression, and anxiety [3,4]. Following the COVID-19 pandemic, 1 in 5 children in the UK (equating to 1.1 million) have reported feeling unhappy with their lives [5], while clinically significant mental health conditions, loneliness, and isolation in childhood increased by 50% [5]. These figures are estimated to be even higher for children from vulnerable groups, such as low-income households, special educational needs/neurodevelopmental differences (SEN/ND), or children exposed to adverse childhood experiences [6]. Another concerning issue is that, although the rate of domestic abuse rose by at least 30%, the number of children coming to the attention of services has fallen dramatically since the first lockdowns and school closures [7]. In addition, the economic uncertainty exacerbated by the pandemic [8] and the environmental uncertainty due to the climate crisis have added significant burdens to children and young people across the globe [9,10]. Children’s mental health needs are now believed to have drastically exceeded the capacities of the NHS and Children & Adolescent Mental Health Services (CAHMS), and there is an urgency to radically rethink the scope of support that children receive [5,8].

Children spend an inordinate amount of time in school, a place where children’s emerging needs can be identified and appropriate and timely support are provided. Schools are often the only place that can facilitate equity of access to mental health services without excluding children who need it the most. Their remit as educational institutions can reduce stigma and increase inclusivity, while also accessing supportive networks of peers, teachers, healthcare professionals, and parents [11,12]. Furthermore, school-based counsellors and psychotherapists can streamline the referral process and target children experiencing barriers due to lack of transportation, parent work schedules, funding, and inadequate treatment from other sources [13].

Nevertheless, a major challenge for school mental health services is that the focus relies heavily on the treatment of severe difficulties or disorders, whereas early detection and prevention may be equally important [2,14]. When opportunities for prevention are missed, chances for school drop-out, self-harm, aggressive or violent behaviour, or even suicide are increased. The centrality of early prevention and intervention has been a consistent theme within governmental policies [15] with a longstanding recognition of the importance of schools in the early identification of children’s mental health difficulties [16,17]. Delayed identification of mental health difficulties in children may culminate in costly crisis interventions, alienation from school, and long-term impacts on children, their families, and communities [17,18]. For example, 7000 children are being excluded from schools annually in the UK (equivalent to 35 children/day), while 1300 of these exclusions come from primary schools [19]. These exclusions could have been avoided with prevention and timely interventions at the early stages of children’s education [20]. In contrast, it is estimated that more than 70% of children lack supportive services at a sufficiently early age [21], 30% of referrals are turned away and waiting lists can take up to a year [19]. Such delays have long-lasting and potentially irreversible negative effects for children’s mental health and well-being.

Despite growing evidence around the impact of arts therapies on children’s mental health and well-being, the central focus on workforce training and intervention implementation in schools has tended toward traditional and empirically recognised approaches such as counselling, Cognitive Behavioural Therapy (CBT), and talking therapies. Arts therapies is an umbrella term referring to art, music, drama, and dance movement therapy; psychotherapeutic approaches that aim to facilitate psychological change and personal growth using arts media. Arts therapies have been defined as, “the creative use of artistic media for non-verbal and/or symbolic communication, within a holding environment, encouraged by a well-defined client-therapist relationship, in order to achieve personal and social therapeutic goals appropriate for the individual” [22] (p. 47). In the UK, arts therapies are recognized professions; art therapy, music therapy, and dramatherapy are regulated by the Health and Care Professions Council (HCPC), while dance movement psychotherapy is regulated by the UK Council for Psychotherapy (UKCP).

Arts therapies have been used widely for children and young people in a wide range of settings, such as hospitals, clinics, and outpatient treatment facilities [23]. More recently, arts therapists have seen a substantial growth of employment in educational settings, bridging the gap between health and education. It is estimated that more than half of all registered arts therapists in the UK are working with children and young people [23], but this may vary for the different types of arts therapies. The latest workforce survey in music therapy [24] showed that schools were the most reported setting, while working with children and adolescents was the most reported post (78% of all posts). The latest workforce survey in art therapy [25] showed that 68% of all art therapies were working with children and young people, while 35% of them were based in schools. In dance movement therapy, it was estimated that 33% of dance movement therapists were working with children and young people, while schools were the third most reported setting (28% of all posts) [26]. Following contact with the British Association of Dramatherapists in 2020, it was estimated that approximately half of the registered dramatherapists were working with children and young people, the majority of whom were based in schools. Despite that, the inclusion and integration of arts therapies into regular mental health provision in educational settings has only recently begun. Underpinning arts therapies with rigorous research will strengthen such an integration [27].

We designed and conducted a pilot cross-over randomised controlled study aiming to (a) explore whether all components of the study (i.e., recruitment, randomization, and follow-up) can work together and run smoothly in a larger trial; and (b) investigate the impact of arts therapies on several quantitative, qualitative, and arts-based outcomes. The protocol was published before the beginning of the study [28]. Furthermore, the first research question (a) was addressed in a separate publication which presented the quantitative evidence of this study [20]. The current article aims to address the second research question (b), presenting the qualitative and arts-based evidence from the children who participated in the arts therapies.

## 2. Materials and Methods

### 2.1. Methodology

Because of the complexity of health difficulties, complex interventions require information from various perspectives and methods [29]. A mixed methodology was adopted in this study, using quantitative, qualitative and arts-based methods to investigate the process and outcomes of arts therapies, both from children’s experiences and with standardized questionnaire-based measures. The mixed methodological approach is philosophically underpinned by pragmatism [30,31], which embraces the positivist/postpositivist and constructive paradigms with the aim to gain a comprehensive understanding of the research phenomenon or problem both from quantitative data (measured facts) and qualitative methods (personal experiences) [32,33]. Epistemologically, there are times in the study when an objective approach is adopted that requires distance from the participants, while at other times the researcher adopts a subjective approach supported by close interaction with participants to understand their realities [34] and produce socially useful knowledge [35]. Therefore, the use of both approaches can offer more breadth, depth, and richness, adding insights that may be otherwise missed [31]. In our study, we took advantage of the strengths of using mixed methods in a complementary way, rather than for cross-validation and triangulation purposes. For example, the aim was not to validate what the children said in the interviews through the standardized questionnaires, but to inform the findings from both approaches and better understand the impact of arts therapies on children’s mental health and well-being.

### 2.2. Study Design

We conducted a pilot cross-over study using a randomised controlled trial design [31] to evaluate the effects of an arts therapies intervention for children in primary schools. Half of the participants were randomly assigned to the arts therapies intervention immediately and then switched to the control group, while the other half acted as a control group in the beginning and received the intervention at a later stage. As such, it was a partial cross-over design. This method has the advantage of reducing the number of participants required, since each participant serves as both a participant in the intervention as well as a control. It also ensured that all children received equal opportunities and benefits from their participation in this study.

As per the National Institute for Health Research [36] (p. 2) definition of pilot studies, the primary aim was to test whether all components of the study (e.g., recruitment, randomization, treatment, follow-up assessments) can work together and run smoothly in a larger trial. Within the pilot, however, certain elements were tested for feasibility [37], particularly: (a) the acceptability of the randomization process to schools; (b) the implementation of the arts therapies protocol; and (c) the methods of process and outcome evaluation.

### 2.3. Study Procedure

Following ethical approval from the Ethics Committee at Edge Hill University, Faculty of Health, Social Care and Medicine, we created a list of primary schools through the public catalogue of public schools in a Northwest region of England. We randomly selected and contacted schools until four schools agreed to participate. The research team and the arts therapists had no prior relationship with any of the schools involved.

The head teachers from these schools were asked to identify two classes having a greater need for psychological support. The teachers of these classes completed the Strengths and Difficulties Questionnaire [38] for each child; this questionnaire was used as a screening tool. Children who were rated as having mild emotional and behavioural difficulties were the targeted participant cohort. The inclusion and exclusion criteria, used at the recruitment stage, are presented in the study protocol [28] and the author’s PhD thesis [39].

An equal number of children were targeted from each school (i.e., 16 children per school). Children were randomly selected through random number generator software and were evenly randomized to the intervention or control groups (i.e., eight children in the intervention and eight in the control group per school). Only the music therapy group had seven children per group, as the music therapist expressed concerns around the noise that would be generated by a larger group of children. In the unusual case where 7/8 participants were of the same gender, the last participant was replaced by a participant of the different gender. For example, if seven girls and only one boy were randomly selected, the seventh girl was replaced by another randomly selected boy. This approach was used to ensure that at least two children of the same gender were allocated in each group, to avoid gender biased findings and outgroup gender isolation. If the selected children (or their parents/legal guardians) declined the invitation to participate in this study, we continued the randomization process until sixty-two children agreed to participate. The recruitment and dropout rates are presented in Table 1.

The parents or legal guardians of these children received the participant information sheet by the school and were invited to a workshop to understand more about arts therapies and the nature of the study. A separate workshop was also delivered to children, which entailed further explanations through child-friendly methods. The arts therapists were present in these informative sessions, offering examples of activities that would take place during the sessions. A different arts therapies intervention was delivered to each school, i.e., either art therapy, music therapy, dance movement therapy, or dramatherapy.

### 2.4. Arts Therapies Intervention

Prior to this study, we conducted two systematic reviews [40,41] which informed the development of the therapeutic protocol based on the evidence of “what works best” for children at schools. The intervention was also influenced by the Arts for the Blues, an evidence-based creative group psychotherapy that was originally developed for adults [42]. A detailed description of the therapeutic ingredients and framework, as well as the activities that took place in each session, are presented in the study protocol [28] and the author’s PhD thesis [39]. Arts therapies sessions were delivered once a week for one hour, over eight consecutive weeks, always at the same day and time.

The recruitment of the arts therapists was made through advertisements in the respective British associations for the four arts therapies. Arts therapists were trained on the protocol application and were encouraged to make their own clinical judgements moderating the structure of the sessions when needed. These modifications were recorded for fidelity purposes. The protocol was used to ensure arts therapists adhered to the overall therapeutic model and to allow future replications. A summary of the therapeutic model is presented in Figure 1.

### 2.5. Methods of Data Collection and Analysis

The process and outcome evaluation were completed through interviews, arts-based methods, standardized questionnaires, and biomarkers (FitBits). Data collection took place before, during and after the arts therapies in the same room as the sessions were conducted but were separate from the sessions themselves. In this paper, only the qualitative and arts-based methods will be presented, as the quantitative methods has been published separately [20].

#### 2.5.1. Interviews with Children

Semi-structured one-to-one interviews were used to understand children’s experiences of participating in arts therapies. All children were interviewed individually one week after the end of arts therapies, for approximately half an hour. Follow-up one-to-one interviews were also conducted at three-, six-, and twelve-months post-intervention to explore any further insights or changes in children’s perspectives. These interviews were shorter and lasted approximately 15 min with each child. All interviews were conducted by the research lead [ZM].

The main themes of interest during the interviews were: (i) what children found helpful or unhelpful, (ii) what they did or did not enjoy; (iii) their most outstanding memories from the sessions; and (iv) whether they practiced any of these activities/techniques in their day-to-day life after the end of arts therapies. To facilitate memory retrieval, the researcher selected and showed 1 or 2 photographs from each session and invited children to share their thoughts and reflections on them.

The interviews were analyzed through reflexive thematic analysis [43] and, as such, the process of coding was fluid and flexible so that codes could evolve and change. Through this open coding process, reflexive thematic analysis allowed us to reflect on how the research team was conceptualizing the data, and how these conceptualizations were evolving, growing, or deepening alongside increased understanding of the data [43].

#### 2.5.2. Children’s Arts Work

The creation of arts work was used as an arts-based method to explore potential changes in children’s emotional expression before and after their involvement in arts therapies [44,45]. During the interviews, children were provided with colours, paper, musical instruments, and materials such as scarfs and fabrics. The same materials were provided to all children, regardless of the type of arts therapies they participated in. They were then invited to close their eyes (if they felt comfortable) and contemplate how they felt at that moment. After taking as much time as they needed, they were invited to demonstrate what this feeling would look like if it was a drawing, movement, gesture, or a sound, using the provided materials.

Hervey’s analytic frame of “open dialoguing” [44] was considered the most appropriate to explore the aesthetic qualities of children’s arts work, especially since some of the younger children were expected to find it difficult to vocalise their feelings. For example, for movements, gestures, or postures, the focus was on children’s effort to express their emotions, whether they used their full body or parts of it, the space they used, or the flow and rhythm. For visual data, emphasis was on aspects such as, colour, background, self-image, or the space that children used on the paper. Similarly, for musical data, the primary focus was on the degree of rhythmic freedom (i.e., limited, unstable, complete), the force beating (i.e., chaotic, creative, emotional) and responsiveness to the mood. A thematic analysis [43] was also used to develop themes around children’s verbal descriptions of their feelings and emotions. In addition, the lead researcher [ZM] included these in the analysis notes from her observations during the process of children’s arts-making.

## 3. Findings

### 3.1. Interviews with Children

Below, we describe the most helpful elements of arts therapies that contributed positively to children’s mental health and well-being, as well as the unhelpful elements which should be addressed in future research and practice.

#### 3.1.1. Helpful Elements of Arts Therapies for Children’s Mental Health and Well-Being

Children expressed that the most helpful elements of the intervention were: self-expression through engagement with the arts; safe spaces; stress relief; empowerment; and the development of coping mechanisms.

##### Engagement with the Arts and Self-Expression

Self-expression through engagement with the arts was one of the most fundamental elements of the changes that children experienced and expressed during the interviews. Children were re-assured from the beginning that all emotions and feelings were welcome in this group. During the interviews, they expressed that the non-verbal communication through the engagement with the arts allowed them to share experiences and feelings that had never been shared before:


*“I was sharing things that I was not used to share with anyone else.”*
(9 years-old-dance movement therapy)

Although drawing was the most familiar medium for children to express both pleasant and unpleasant emotions and experiences, familiarity with other types of arts was developed over time. For example, storytelling and story enacting offered children “permission” to act in any manner they preferred without being judged, which is potentially why they showed a strong preference towards enacting “crazy” or “silly” characters (Figure 2a). A child said:


*“I made it [the character] look like crazy, you know, like my auntie used to be before she passed away. I want to be like my auntie so this [character] was actually me.”*
(9 years-old-art therapy)

They also enjoyed the process of choosing their own character(s), a choice they rarely had in school plays and performances. Children especially enjoyed the stories that allowed characters with different energy levels to emerge and co-exist. For example, during a role-play, some children pretended that they were pirates on a boat, while others were relaxing on a peaceful island. Children had the opportunity to change roles when they wanted, giving them the time they needed to remain still or active:


*“One of my favourite [role-play] was the pirates on the island, where we could choose to fight with the pirates on the boat if we wanted, or stay still on the island.”*
(9 years-old-dance movement therapy)

Puppetry was also used as a metaphor and medium to allow children to talk about difficult things they would not have otherwise shared. For example, in a dramatherapy session, children were invited to enact how “a difficult day feels like” and communicate it to others non-verbally through their puppets. In the interviews, most children said this was one of their favourite sessions because it gave them the chance to share their own stories, but also keep them private as nobody knew whether they were real or not:


*“I could tell my own story through my own puppet, but nobody knows if it’s a true story or if I made it all up.”*
(8 years-old-dramatherapy)

Similarly, in music therapy, self-expression was facilitated by music making and song writing. Children shared that their chosen lyrics were related to their real life, and they were curious to hear about other children’s life experiences through their songs.

Regardless of the type of arts that children chose to engage with, materials that enabled them to use their senses facilitated emotional expression and made a strong impression on children who could describe, even a year later, how it felt smelling the clay, listening to the sound of an unusual instrument, or touching the sand (Figure 2b):


*“I placed all my family in my sand tray, some trees around and a lake that I go with my parents to relax and spend the weekend sometimes […] the sense of the sand was very relaxing, reminded me of our weekends away.”*
(10 years-old-art therapy)

##### Safe Space

In all arts therapies, children had the opportunity to reflect on the importance of having a safe space and how it feels to be safe. In art therapy, children made their own “safe bench” (Figure 3a), in dramatherapy they made birdhouses that make birds feel safe (with “birds” as a metaphor for their own self), in dance movement therapy they made their “safe home” with scarfs, blankets, and other materials (Figure 3b), while in music therapy children explored the sounds that made them feel safe or unsafe.

Another element that contributed to children feeling safe in the sessions was the awareness of the importance of confidentiality, which was one of the key ground rules established from the first session. All children agreed that what was being said in the sessions had to stay in the group and not be shared with people outside the group. This offered children with a sense of responsibility the means by which to protect other members of the team:


*“We all know that it’s not like gossip what we say and we have to protect each other.”*
(9 years-old-music therapy)

Some children expressed how they started to treat other conversations with their friends or families as confidential, which appeared to result in developing more trusting relationships and safety in sharing personal experiences with others:


*“It was easy to share when I choose the people I want to share with and I know that we all do the same […] confidentiality was very helpful.”*
(9 years-old-dance movement therapy)

Some groups started by dancing and singing to the “Confidentiality Rap”, which was a symbolic activity to create a clear boundary between the class and the arts therapies sessions. Similarly, ending the sessions with the same song or activity gave children time to prepare for the transition back to class:


*“I liked the sessions that started and finished with the same song to know that I am not in the class anymore, and to prepare me to go back to the class. The song in the end was my favourite; it made me happier and ready to go back to class.”*
(8 years-old-music therapy)

##### Stress Relief

Some children were experiencing stress that affected their day-to-day life, as well as their performance at school. Almost all sessions included activities with relaxation techniques, such as deep breathing, which were beneficial especially for these children:


*“When I feel upset I focus on my breath and it helps me calm down.”*
(9 years-old-dance movement therapy)


*“It made feel less stressed than I was used to be. I am still stressed in the inside but not on the outside anymore. Before I would be stressed in the inside and also the outside.”*
(10 years-old-art therapy)

Some children shared that the calmness they experienced during the sessions was transferrable to the rest of their day and enabled them to have better quality of sleep:


*“It made me sleep better because I felt less stressed.”*
(10 years-old-dramatherapy)


*“I try to relax for ten minutes before I go to bed and I sleep better than before.”*
(9 years-old-dance movement therapy)

Several children said they maintained this practice even three months later.

##### Empowerment

Children mentioned that the sessions that involved empowerment activities improved their self-esteem and self-confidence. This also became apparent through the arts-based methods, as described below. The empowerment activities that children found the most helpful was creating their dreamcatchers, exploring their “superpowers” and making their “boat of difficulties”. In the “dreamcatchers” activity, children reflected on their dreams and hopes for the future, and how they can achieve them. One child said:


*“My dream is to become a head teacher, that’s the wish I wrote down on my dreamcatcher and I hope this dream will come true one day […] to help other children feel good, take good grades, have fun at school and be fair.”*
(10 years-old-art therapy)

Pretending to be “superheroes” and reflecting on their “superpowers” made children feel “*special*” and “*confident*”. When some children found it difficult to think of any “superpower”, the rest of the team would help them realise what their strengths are, which also facilitated team building and collaboration between the group. As one child shared:


*“I had a vision of myself as a volcano which has superpowers. When I’m angry, I look like the fire. When I’m calm, I look like the water. When I’m in the middle, I look like the clouds. I notice non-stop different things in my mind; water, fire, clouds… but most of the time I feel like the fire.”*
(10 years-old-art therapy)

In the “boats of difficulties” activity, children were provided with materials to make their own boats and were invited to put difficulties inside that they were dealing with, as well as past experiences or memories that they did not want to keep. All boats eventually came together and sailed away. A child expressed that “*it was a relief*” seeing his difficulties sailing away, but also that he realised that “*everyone has difficulties*”. Another child mentioned that:


*“When we were sharing the boat of our difficulties, I was thinking that things will get better and I felt better.”*
(9 years-old-art therapy)

##### Coping Mechanisms

Children referred to several coping mechanisms, with the most common being the development of patience, active listening, and the gradual control over the behavioural reactions that arise from uncomfortable situations. For example, one child acknowledged the lack of listening skills and sense of control over his feelings prior to the arts therapies:


*“I used to be really angry, I wanted to destroy everything. I had never listened in my life, never. I couldn’t even concentrate.”*
(10 years-old-art therapy)

Another child noted that as she became more patient, she experienced less conflicts with her siblings and they were able to develop a warmer relationship:


*“I’m fighting less with my brothers, and I control my anger. I try not to respond immediately when something happens and take my own time when I need it.”*
(10 years-old-dance movement therapy)

There were also indications that some children started taking ownership over their emotions, behaviours, and actions:


*“When I am frustrated, I know that there are better ways than being mad at other people when they haven’t really done anything and it’s not their fault.”*
(9 years-old-dramatherapy)

Perspective was another coping skill discussed by children. For example, some mentioned that arts therapies helped them to zoom out of what was not going well in their life, keeping a wider perspective and focusing on what was going well:


*“I got to forget about all the bad things, or most of the bad things in my mind.”*
(9 years-old-dramatherapy)

Finally, some children mentioned that this sense of perspective and appreciation of the peer support they received during the sessions changed their attitude towards school:


*“I feel happier coming to school.”*
(7 years-old-music therapy)

#### 3.1.2. Unhelpful Elements of Arts Therapies for Children’s Mental Health and Well-Being

The most unhelpful elements of the intervention that children expressed were a lack of time for tasks that they enjoyed; a lack of time to build group cohesion in groups in which it was lacking; and a small number of sessions, as well as a large number of children per group.

##### Lack of Time

Children expressed that they preferred to have less activities with adequate time to fully focus on each one, rather than rushing quickly from one activity to the other.


*“I could spend hours, no, I could spend days in each craft we did. I didn’t want to feel in a rush.”*
(8 years-old-art therapy)

However, they also mentioned that they did not want to spend the entire time on the same activity, because they felt that they were potentially missing out from other creative activities. For example, in music therapy, children would have liked to spend more time playing music and less time in the opening and closing group discussions. In dance movement therapy, children mentioned that, although they enjoyed the relaxation activities, they did not want it to last for too long because they were falling asleep and found it challenging going back to the class. Having one brief opening activity, one main and longer activity, and one brief closing activity appeared to be the ideal timeframe for most children to take advantage of the benefits of each activity.

##### Lack of Group Cohesion

On several occasions, the randomization led to creating groups with members that did not get along well with each other. When this was the case, children were unwilling to discuss or collaborate with each other, to resolve their conflicts, and to openly share with the rest of the group:


*“I didn’t feel comfortable to share around S, we fight a lot in the class and I can’t concentrate when he is around.”*
(10 years-old-art therapy)

Even if their relationships were getting better over time, there were concerns as to whether things were actually resolved affected the children and their participation:


*“It was when we did the mirroring that D said “I don’t want to do this, I don’t like that” but I think he just didn’t want to be a partner with me […] I didn’t feel good about it.”*
(9 years-old-dance movement therapy)

Even though the arts therapists contained the group as a whole and held separate discussions with some children when needed, these circumstances affected the dynamic of the rest of the team and their experiences in the group:


*“It was so annoying and frustrating when R and E were shouting at each other so loud that I couldn’t focus for the rest of the time.”*
(10 years-old-dramatherapy)

Based on children’s feedback, both the number of sessions and the number of children in each group should have been different, as described below.

##### Small Number of Sessions and High Number of Children

Eight sessions of arts therapies was a short amount of time for most children. At the peak of connection between the members of the group, the sessions were approaching the end. Several children expressed during the interviews that they wished to be better prepared for the end of the sessions, with weekly reminders right from the start. As some children expressed:


*“I felt very sad because the sessions were over. I was crying without reason.”*
(9 years-old-dance movement therapy)


*“I felt upset because I didn’t want to let it go.”*
(7 years-old-music therapy)

Some children also expressed that, if each group had less children, they would have taken better advantage of the benefits that each session had to offer. They would have more time to work individually, as well as in a team, and they would potentially have time for more activities.


*“I would prefer smaller numbers [of children], like five of us in each group.”*
(8 years-old-dance movement therapy)

Furthermore, the arts therapist’s attention would not have to be divided between eight children. Considering the randomization method and the grouping of children who did not get along with each other, smaller groups would have made it more manageable for the arts therapists to contain the group and to facilitate the resolution of any tensions arising.

### 3.2. Children’s Arts Work

#### 3.2.1. Themes of Children’s Arts Work

Significant changes were observed in children’s emotional expression before and after arts therapies. Children expressed a wider range of emotions, and they appeared to be more comfortable sharing them not only visually through drawings, but also through sounds, movements, or gestures.

Prior to the intervention, the most common emotions and feelings that children shared were: (i) happiness (28 children); (ii) confusion (13 children); (iii) connectedness with people (9 children); and (iii) sadness (4 children). Seven out of 62 children preferred not to say how they felt.

Post-intervention arts-based data were collected by 56 (instead of 62) children because of six dropouts (Table 1). The most common emotions and feelings were: (i) happiness (23 children); (ii) mixed emotions (14 children); (iii) connectedness with people and places (8 children); (iv) excitement (6 children); (v) confidence (4 children); and (vi) calmness (3 children).

It was striking that, pre-intervention, 13 children shared that they felt confused or that they did not know how they felt. Post-intervention, however, this sense of confusion was replaced by an increased acknowledgment by 14 children that people can have mixed emotions and feelings that occur simultaneously. For example, many children mentioned that they felt “happy and tired”, “happy and sad”, “happy and excited and sleepy”, or “so and so”. One child added that “There is a little happiness in every anger, and a little anger in every happiness”. As such, some children became less reluctant to own emotions that are often considered as negative, such as anger or sadness. For example, while pre-intervention all children who expressed sadness drew someone else (usually the person that makes them feel sad), post-intervention, all children drew themselves instead. Children appeared to have gained confidence in owning different kinds of emotions and accepting all feelings as normal.

Another noticeable difference was in the theme of “connectedness with people”; while pre-intervention most children drew family members, post-intervention they drew members of their arts therapy group. In fact, almost all drawings represented at least one group member, suggesting that some children had developed a sense of belonging, as well as trusting and supportive relationships with other team members.

#### 3.2.2. Aesthetic Qualities of Children’s Arts Work

Most children at the pre-intervention stage were very reluctant to use any of the provided musical instruments and materials, apart from the crayons, markers, and papers, potentially because children were more familiar with them. In addition, children did not know the researcher [ZM] at this time; therefore, sharing may have been challenging or even intimidating. In contrast, post-intervention, children used all the materials provided, including musical instruments, scarves, and fabrics. For example, one child wrapped a scarf around her body to express that she felt “calm and comfortable with myself” and that “this is the calmness that comes with satisfaction in life”.

While pre-intervention all children chose to remain in the same position without moving around, many children post-intervention made use of the space confidently to communicate their emotions more clearly. Similar trends were observed in drawings; pre-intervention, children drew themselves very small, occupying only a small part of the paper, and usually being surrounded by others. However, post-intervention, they drew themselves bigger and taking most of the space on the paper or being on their own. In most drawings, there were details on faces and facial expressions; in some drawings, for example, a smile was clearly marked and was sometimes bigger than the whole face. When the full body was illustrated in the drawings, the extension of the hands or other parts of the body were used to show the extent of that specific feeling; for example, long hands widely open were used to communicate the extent of happiness that children experience. These explanations were provided verbally by the children themselves.

Regarding children’s drawings, the increased use of colours, the complexity of the characters, as well as the addition of things in the background to allow for a better understanding of the context, were noticeable. The metaphoric use of objects and symbols for self-expression was also observed, particularly in the post-intervention drawings of children who participated in the dramatherapy groups.

In terms of movements and gestures, children put a significant effort in expressing their emotions using their body, projecting less reluctance and more confidence. While pre-intervention children used mostly their facial expressions, post-intervention they used their whole body to show the degree or intensity of each feeling. Post-intervention children displayed higher intensity in their movements. For example, children expressed their confidence through raising their chests out of the rest of their body, extending the chin out of the face, and with their eyes looking up toward the sky. Making use of a strong weight and an increased sense of verticality, children rendered the classic stance of a “superhero” to show that they felt “confident”, “strong”, and “cool”. It was also noticeable as to how children’s synchronisation and harmony between emotions and movements had been significantly improved. Pre-intervention, children’s movements or gestures lacked flow and were often not aligned to the emotions they were sharing. For example, verbal expressions of “happiness” were not visible or in harmony with their movements and facial expressions. Post-intervention, however, there was a clear flow and rhythm between the movement of the head, the arms, and the entire body, which aligned with the emotions that children were trying to convey. As such, verbal and non-verbal expressions appeared to be in harmony.

Similar changes were observed with children’s sounds and musical productions. Children appeared to be moving from limited or unstable rhythmic freedom at the pre-intervention stage, towards a great extent of being or looking confident with this freedom. They moved from producing sounds chaotically toward a rather refined rhythm-making which mirrored their specific emotion. For example, drum beating became more emotional, calm, or cheerful, depending on how each child experienced the emotion of “happiness” and the way they preferred to convey it. Furthermore, children seemed to be moving towards rhythmic freedom as they did not hesitate to experiment with new rhythms and sounds and were not concerned about how the final product might sound.

## 4. Discussion

### 4.1. Overview of Findings

Children’s interviews and arts work suggested that arts therapies facilitated self-expression, provided a safe space, empowered children, and supported the development of coping mechanisms. Engagement with the arts as a coping mechanism under difficult circumstances was highlighted, particularly in terms of expressing emotions and feelings which are complex and cannot be easily verbalised. These findings echo existing evidence on the importance of engagement with the arts as a non-verbal communication medium and crucial therapeutic element for children’s mental health and well-being [22,46,47,48,49].

Another significant change was that children started accepting some challenging feelings and emotions as normal, such as anger or fear. This led not only to higher self-acceptance, but also to higher acceptance and understanding of others. The arts-based data also suggested that children became more conscious of the complexity of their emotions, for example that conflicting emotions (e.g., happiness and sadness) can often co-exist. Drawing upon humanistic and person-centred theories, safely accessing and expressing feelings can facilitate self-awareness and have cathartic effects [50,51]. Furthermore, some children were gradually taking more ownership of their emotions, behaviours and actions, and developed an increased sense of self-control; both of which are key elements of the empowerment theory [52].

Children reflected on the positive feelings that they experienced during the arts therapies. Specifically, they mentioned that the sessions made them feel happier, safer, calmer, and that they enjoyed the sense of togetherness and belonging in the group. This is particularly important considering the distractions and interruptions that most groups faced. This suggests that, even though it might be challenging to ensure privacy in schools, a safe therapeutic environment and relationship can still be achieved with a mutual effort from the arts therapists, the school, and the participants.

The above findings link to the PERMA theory [53], according to which, the elements of Positive emotions, Engagement, Relationships, Meaning and Accomplishment act as enablers of well-being. Based on children’s experiences and perspectives, it became explicit that arts therapies can enable all these elements. The findings also link to the self-determination and self-actualisation theory [54] according to which, well-being is dependent on the fulfilment of three core needs: agency, mastery, and relatedness. When these needs are met, children with these attributes can feel confident and assured in their ability to achieve their aspirations and fulfil their own potential (what is known as self-actualisation), whilst also maintaining positive and healthy relationships with others. As children said, arts therapies addressed all of these core needs.

There were, however, aspects that prevented children from getting all benefits that arts therapies have to offer. These aspects were spending too much or too little time on the same activity, lack of group cohesion, the small number of sessions and the high number of children per group. Below we provide some recommendations that could address these limitations in future research and practice.

### 4.2. Improving the Quality of Group Arts Therapies

Despite the advantages of providing arts therapies in school settings, there are inherent barriers that impede to the therapeutic work. For example, finding a private and safe space proved to be exceptionally difficult. The feasibility of privacy and confidentiality within educational environments has been frequently questioned [55], while also the school calendar is filled with holidays, school trips, ceremonies and activities that interfere with the flow of arts therapies. Most schools could not provide us with rooms appropriate for therapeutic use, an issue that has been commonly reported in other studies [56]. As a result, the sessions took take place in rooms intended for different purposes; for example, the dramatherapy sessions took place in a storeroom. This does not mean that schools are unsuitable spaces for therapeutic work, but the barriers need to be acknowledged and addressed insofar as possible. Although research suggests that there is a positive correlation between the arts therapists’ satisfaction with the suitability of the therapeutic environment and the clients’ outcome measures, this correlation might not be statistically significant [56], therefore the limitations in the therapeutic space might not significantly affect the therapeutic outcomes. The strategies below enabled us to improve the privacy and safety of the therapeutic environment and are strongly recommended.

The first recommendation is to deliver an arts therapies workshop to the parents or legal guardians and to the school staff prior to the beginning of the sessions. We did not deliver this workshop with the first cohort, but it became clear that the importance of a safe therapeutic environment was not fully understood; as a result, there were frequent interruptions during the sessions that interfered with the therapeutic process. When we delivered this workshop with the second cohort, we noticed a significant change in the school staff’s attitudes towards protecting the therapeutic space and not interrupting unless in case of emergency.

The second recommendation is setting clear boundaries between the beginning and end of arts therapies through consistent rituals that provide children with time to transition from an educational to therapeutic environment, and vice versa. All sessions started and ended in the same way; a song, dance, movement or drawing that was a sign of welcoming children in the group and ended with a similar activity as a closure. This method was also applied when interruptions or distractions occurred, as children needed again time to re-engage with the therapeutic process. Children expressed that this routine was important for their transition in and out of the therapeutic space and to prepare them for going back to class.

The third recommendation is to allow sufficient time for trusting relationships to evolve, particularly between the research team, the arts therapists, and the school staff. At the beginning of the intervention, some arts therapists were concerned that the school staff did not understand the importance of their work and did not protect the therapeutic space. However, even within a couple of weeks, the relationship between the arts therapists and school staff had evolved and made it easier to communicate each other’s needs. For this reason, whenever possible, it is recommended that the interventions take place at schools which are already working or have worked in the past with arts therapists. Building longer collaborations may be the key in protecting therapeutic space. Regev et al. [55] also suggested that longer collaboration between arts therapists and teachers is the key for maximum impact of school-based arts therapies. Specifically, they found that arts therapists who collaborate with schools for long periods of time are more likely to have their own therapeutic space, and for the schools to invest in the materials and resources that are needed [55]. Moreover, a synthesis of trials in educational settings [57] found that the establishment of rapport and positive relationships between the research team and the school staff was among the most important retention facilitators that determined the feasibility of trials in schools.

Children recommended that there should be no more than six children per group, so that everyone can receive sufficient attention by the arts therapist, and more time for sharing. They also recommended that two more sessions would have been beneficial, and they would have liked to have weekly reminders about the remaining sessions right from the beginning, so that they can be mentally prepared for the sessions to reach their endpoint.

Most importantly, the randomization process led to shaping groups with members who did not get along well with each other, or their energy level was conflicting. This led to arguments that were challenging to resolve and reluctancy to share personal experiences and thoughts from the whole group. As a result, the group cohesion (or lack thereof) impacted significantly on both the therapeutic process and the therapeutic outcomes. A strategy worth considering in future experimental studies is seeking advice from the teachers as to how children should be clustered into groups following the randomization. This method could help to bring together children who get along with each other, protecting the therapeutic environment for the group as a whole. Alternatively, a pre-intervention group assessment would be crucial to determine whether the group has an adequate level of group cohesion. Finally, another solution could be the delivery of a one-to-one session with each child before their allocation into groups to understand whether some children would find it difficult working in a group, or whether they can handle sharing the arts therapist’s attention. Although this strategy is not cost-effective, it could provide valuable insights as to whether groups are an appropriate form of therapy for some children and to minimise the risk of harm for the rest of the group.

### 4.3. Strenghts and Limitations

This pilot was developed based on two systematic reviews [40,41] which synthesized previous interventions that had been successfully implemented and assessed by children themselves, and it was informed by other evidence-based arts therapies protocols [42,58,59]. The therapeutic protocol and intervention were evaluated by children, arts therapists and the research team [20] and is expected to inform the development of future interventions in school-based arts therapies. The detailed description of the therapeutic protocol and its principles is expected to make this intervention replicable in future studies.

The implementation of quantitative, qualitative, and arts-based data helped to gain an in-depth understanding of the outcomes and process of arts therapies that comes purely from children’s perspectives. To reduce power imbalances and misinterpretations of children’s views, all tentative interpretations from the qualitative data were made available to children for member cross-checking. This helped to ensure that the findings represented children’s viewpoints, contributing to the improvement of credibility and accuracy of the findings [60,61].

Despite the member cross-checking approach, it is possible that some children might have been hesitant to openly share whether they agreed with the initial interpretations and findings, or what they did not enjoy or find helpful about the intervention. In addition, cross-checking the findings from children’s arts work was a rather perplexed process and the interpretation stems primarily by the researcher [ZM]. Therefore, there are limitations to the reliability of the arts-based findings. Since the intervention lasted approximately three months, and that the lead researcher [ZM] was present in every session as a participant observer, the relationship with the children became closer over time. This is potentially why we received more feedback on what worked well compared to what did not work well in this intervention.

## 5. Conclusions

Children verbally and artistically expressed that they experienced positive changes in their mental health and well-being, such as self-expression, safety, empowerment, hope, and optimism for the future. The arts were particularly important for expressing complex emotions and feelings that cannot be easily verbalised. These benefits were linked to humanistic theories [49,50], self-determination and self-actualisation theory [53], PERMA theory [52], and empowerment theory [51]. This study employed a novel approach to working with children, embracing all arts therapies as one research domain and setting children’s verbal and non-verbal responses at the heart of outcome evaluation. This study also highlighted areas for improvement based on evidence grounded on children’s perspectives. Redirecting the focus of research to encompass children’s perspectives may result in better-informed policies and practices, encouraging decisions that are aligned to children’s needs. The implementation of the recommendations discussed in this article may increase the benefits for children’s health and well-being, as well as the wider recognition and inclusion of arts therapies in national and international health-related guidelines. This may be a crucial step for the survival and thriving of arts therapies in educational and healthcare systems worldwide.

## Figures and Tables

**Figure 1 children-09-00890-f001:**
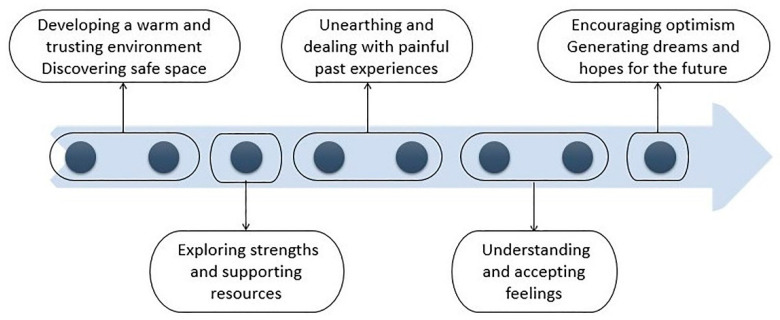
The therapeutic model followed in the arts therapies intervention protocol.

**Figure 2 children-09-00890-f002:**
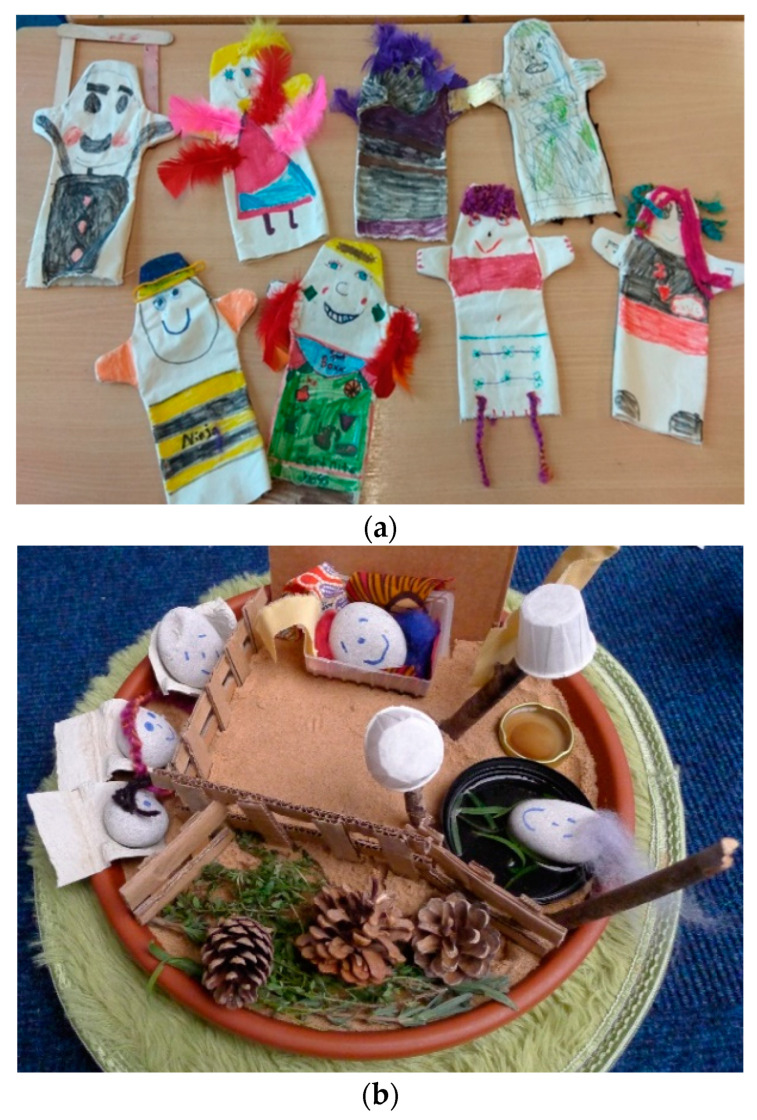
Engagement with the arts and self-expression. (**a**) Puppets—dramatherapy; (**b**) Sand trays—art therapy.

**Figure 3 children-09-00890-f003:**
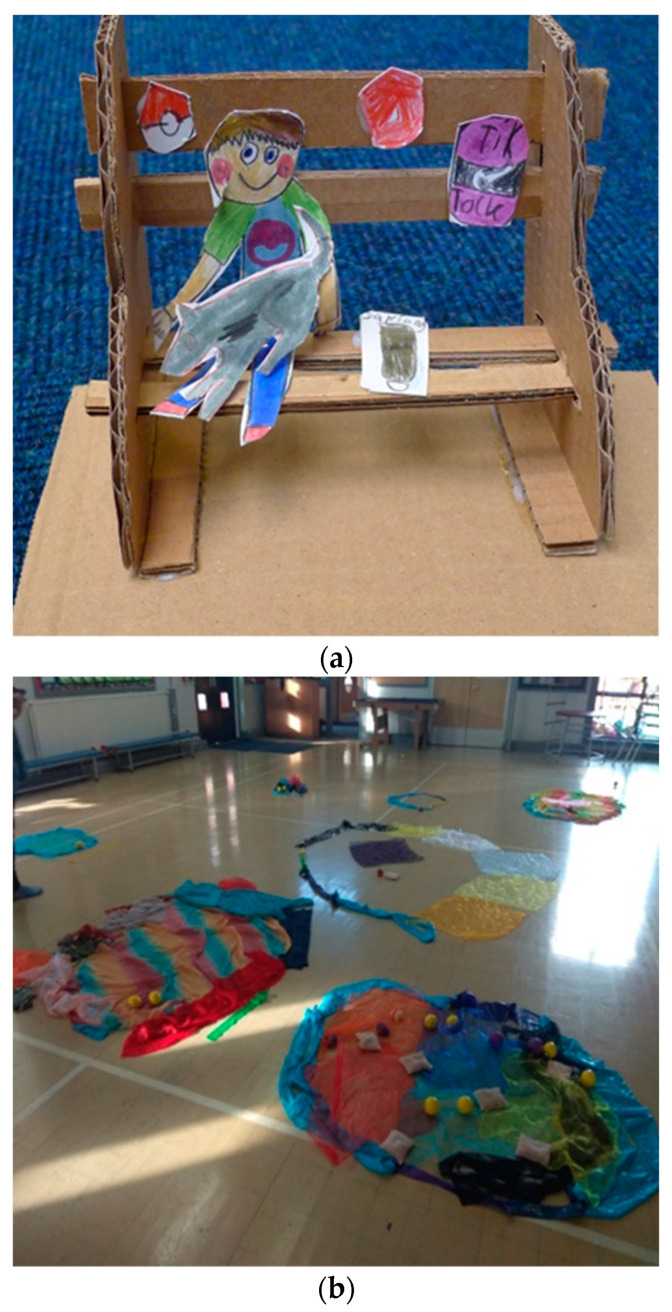
A safe space. (**a**) “Safe bench”—art therapy; (**b**) “Safe home’”—dance movement therapy.

**Table 1 children-09-00890-t001:** The recruitment and dropout rates (with reasons) for each type of arts therapy.

Type of Arts Therapies	No. of Children Recruited	No. of Children Completed	Dropouts (Group)	Dropout Reason
Art therapy	16	14	2 (control group)	1 hospitalisation 1 left school
Dramatherapy	16	14	2 (1 from each group)	2 did not want to keep missing the PE classes
Dance movement therapy	16	16	N/A	N/A
Music therapy	14	12	2 (intervention group)	1 left school 1 did not want to keep missing the assemblies
Total	62	56	6	

## Data Availability

Data supporting reported results can be found in the author’s Open Access PhD thesis: Child-focused process and outcome evaluation of arts therapies for children in primary schools—Edge Hill University.

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
