# Peer review of "Qualitative and Arts-Based Evidence from Children Participating in a Pilot Randomised Controlled Study of School-Based Arts Therapiesâ€"

_children, 2022, doi:10.3390/children9060890_

Round 1
Reviewer 1 Report
The article describes the perspectives of children on their experiences in therapy. This is an important article, which presents a point of view that is rarely described in the art therapy literature, mainly that of children in treatment.
My comments are as follows:
Abstract: The abstract is not accurate enough, mainly with regard to the method. The phrase “randomized controlled study research” is confusing in relation to the part of the study presented in this article. There is no reference in the abstract to the research instruments which makes it difficult to understand
Introduction: The introductory section provides a comprehensive and well-written overview of the importance of providing a therapeutic response to children in school, and art therapy in school settings. At the same time, the introduction, as it currently stands, applies to multiple types of research questions, and the literature review fails to focus on the research question. There is a growing body of literature about art therapy in schools, but almost none test effectiveness, and children's perspectives have hardly been discussed at all. You may want to use Harpazi's article (https://www.frontiersin.org/articles/10.3389/fpsyg.2020.518304/full) to center on the innovativeness of this study.
It is important to note, as you did, that this study is part of a broader project. However, it is important at the same time to give the reader a complete and focused picture of the current work. Thus, mentioning research questions in the concluding paragraph of the introduction that are not part of the present study is inappropriate.
Method: Continuing from the previous comment, it is inconsistent, as I see it, to describe the study as a mixed method design if the manuscript is only about the qualitative part.. I suggest deleting the references to aspects of the larger project that are not reflected in this study from the method. The broader study can be summarized in 2-3 sentences at the beginning of the method section before dealing exclusively with the current study. The recurring references to a randomized control trial, and a pilot study, that have nothing to do with qualitative approaches is particularly confusing. They should simply be mentioned when summarizing the main drivers of the larger project.
The purpose of the intervention is shown in Figure 1, but is not described in the text. I suggest adding a description of the purpose of the intervention.
There is no procedure section, and not enough information about when the interviews with the children took place, how many interviews were conducte , and by whom. The findings suggest indirectly that some of them took place over a long period of time, at least three months, or one year, and that the first author served as the interviewer, but this should be noted explicitly in the method section.
Results: The findings are presented in a clear and interesting way overall. Some of the titles of the themes do not indicate their content; for example, Number of sessions and number of children, please correct this. Some of the findings are difficult to understand without knowing when the interviews took place.
Discussion: Please check and correct statements that have over-generalization such as “Another significant change was that children started accepting all feelings and emotions as normal” (lines 581-582). It would be more accurate to formulate this in a less inclusive way.
On lines 618-619 there is a reference to Articles 56 and 57, but to the best of my knowledge they are irrelevant to the statement attributed to them in these lines. Please check that there are no errors in other references as well.
It is important to make sure that the discussion deals with the findings presented in the findings section. Again, it is confusing to read about the findings from the broader project. For example, lines 629-636 discuss the parents’ preparation for the intervention, but it is not clear how this relates to the findings.
In the discussion as well as in the introduction and method sections, the author should focus on the current study and not on the broader study. For example, line 693 discusses quantitative tools that have nothing to do with the study presented here.
Author Response
Dear editor,
We would like to thank both reviewers for their time to review this article and for giving us constructive feedback. Please see below all the amendments we have made to the updated version:
As per the reviewers’ suggestion, we have rephrased the title to emphasise that this study was focused on children’s reflections/perspectives.
We have now added in the abstract the methods of data collection that we used in this study (i.e., individual interviews with 62 children after the end of arts therapies) as well as the age of the children.
There were conflicting views in the reviews we received, with one suggesting that we shouldn’t be referring to the quantitative results/article at all, and the other suggesting that we should mention more about this article. In the light of this, we thought it is important to keep in the article the research questions for all the study, but we have made more explicit which research questions we are addressing in this article, and which research question we have addressed in the quantitative article.
In terms of the methodology, we thought that it is essential to discuss about pragmatism as this was our ontological and epistemological positioning. Even though this article presents only the qualitative evidence, our philosophical positioning remains the same. One of the reviewers also mentioned that ‘references to a randomized control trial have nothing to do with qualitative approaches’ – in fact, we are trying to question/challenge this way of thinking and we strongly believe that experimental studies and qualitative methods should go hand in hand. See for example the richness of information we were able to gather through the qualitative methods even though it was an experimental study.
The first reviewer also mentioned that there is no procedure section but there is a study procedure section starting from line 154. However, we have now added more information with regards to when the interviews took place, how many interviews were conducted, and by whom.
The second reviewer also asked how long the sessions lasted for, this information is provided in line 201: “Arts therapies sessions were delivered once a week for one hour, over eight consecutive weeks, always at the same day and time.”
In the results section, we have rephrased some of the titles / subtitles to give a better idea of the content, as the first reviewer suggested.
In the discussion section, we have included more literature, such as Harpazi's et al (2020) article on Perceptions of Art Therapy in Adolescent Clients Treated Within the School System. We have rephrased over-generalizations, such as in lines 587. We have also doubled checked all references for errors and have corrected those. A paragraph related to the quantitative results has been removed as suggested as well.
In the limitations section, we have highlighted the role of the researcher as participant observer who was present in every session, and the impact this may have on the findings.
We hope that the above amendments address the reviewers’ comments, but please let us know if additional amendments are needed.
Reviewer 2 Report
The article is relevant, the methodology is clever and thorough. As far as I can judge from a non-English speaking point of view, it is well well-written, concise, and coherent. From the point of view of an academic, a creative arts therapist in special education and a group facilitator I can only offer the following issues and questions:
- This thorough research has been separated into several articles due to its complexity and the richness of data however, the separation is far too hermetic; having a look at the quantitative article, I would suggest mentioning its content a bit, for context.
2. the reader can wonder from time to time whether it is really about the efficiency of the method or about the children's reflections: the title, keywords, abstract and introduction lead strongly towards a methodological review, but the results section strongly focus on the children's reflections. Something should be said both about bot issues, but the results section needs to suit the title: If the subject is the study's efficiency, the results should focus on this issue. If the article describes the children's reflections, the following manuscript covers them well, but then consider changing the title, keywords, abstract and introduction accordingly. You can still relate to important point of methodical efficiency in the concluding part.
3. Also, for context, mentioning the participating children's ages or class in the beginning (abstract and on) not only in the results section, can be helpful.
4. Additionally, how long was each group session? It may be essential for understanding the results and the concluding remarks on the efficiency of the research method.
5. Regarding group work as part of the methodological considerations:
· As a music therapist at schools, I don't understand the difference between 7-8 participants. I think this explanation is a bit confusing and it may be redundant.
· Regarding the need to reduce the number of children in each group, diversity (boys-girls), aggression in groups, as well as the time of each session and the number of facilitators per group, it is advisable to rely on some group therapy literature, not only the research results. Such literature already offers experienced view regarding arts therapies and children's groups.
· The children's reflections on the length of sessions, the process, and the size of the groups can be practical, but it is worth mentioning that due to their age and role as participants in therapy, these reflections are also biased. Please consider finding a way to show the possible bias.
6. It is not clear enough if the writer/researcher was present at the meetings, what practically did she do, how exactly the children experienced her during this process, and what can be the possible effects of her involvement on the results.
Other than that, it is an important research and article. One cannot underestimate its importance for the children's health not only locally, but internationally. Thank you!
Author Response

(The authors gave the same response as above.)
